# Study protocol for an observational research study with an embedded N-of-1 design: Increasing the availability of goal-oriented cognitive rehabilitation for people living with dementia in Ireland

**Caoimhe Hannigan[1], Antoine Lemercier**  **[1], Garret McDermott[2,3], Helena Lydon** [4]*****,
**Sean P. Kennelly[3,5], Michelle Kelly[1]**

**1** Department of Psychology, School of Business, National College of Ireland, Dublin , Ireland,
**2** Department of Psychology and Regional Specialist Memory Clinic, Tallaght University Hospital, Tallaght, Dublin, Ireland, **3** Institute of Memory and Cognition, Tallaght University Hospital, Dublin, Ireland, **4** School of Psychology, University of Galway, Galway, Ireland, **5** Department of Medical Gerontology, School of Medicine, Trinity College Dublin, Ireland

* Helena.Lydon@universityofgalway.ie

## Abstract

### Background

Goal-Oriented Cognitive Rehabilitation (GREAT-CR) is an evidence-based early intervention for people with mild-moderate dementia that improves goal attainment and functional outcomes. In Ireland, this intervention is not widely available, likely due to resource shortages within memory services. This project aims to pilot and evaluate a new service delivery model for GREAT-CR, and to assess its impact on goal attainment, cognition, and quality of life for people living with dementia.

### Methods

This pilot study implements a new service-delivery model where trainee psychologists will offer GREAT-CR within Memory Clinics in Ireland. Sub-study 1 will evaluate the impact of GREAT-CR in this context and sub-study 2 will evaluate the implementation model. Sub-study 1 will employ a quantitative pre-post design with an embedded randomized N-of-1 experimental design. Outcomes will include goal attainment, quality of life, and cognitive function. A subset of three participants will be randomly selected for an N-of-1, single-case experimental design evaluation. The N-of-1 design will include within-case and start-point randomization. Sub-study 2 will include a mixed-methods approach to evaluate the feasibility, acceptability, appropriateness, and potential sustainability of the implementation model at organizational, practitioner, and service-user levels. We will conduct semi-structured interviews with key stakeholders and analyze the data using framework analysis. Quantitative implementation outcomes will be assessed against predefined targets to support qualitative findings.

**Data availability statement:** No datasets were generated or analysed during the current study. All relevant data from this study will be made available upon study completion.

**Funding:** The current research is funded by the Alzheimer Society of Ireland under the Dementia Awards Scheme. The funders had no role in study design, data collection and analysis, decision to publish, or preparation of the manuscript.

**Competing interests:** The authors have declared that no competing interests exist.

## Discussion

The results will inform guidelines for wider service delivery. This project aims to ensure immediate and long-term impacts for participants with early-stage dementia and their carers by increasing opportunities to access evidence-based psychosocial interventions; for trainee psychologists by creating novel placement opportunities; and for service providers by disseminating GREAT-CR implementation recommendations. By addressing current staffing and resource limitations, this model has the potential to enhance the availability of GREAT-CR for dementia patients in Ireland.

## Background

People living with dementia (PLwD) can retain their abilities, learn new skills, and improve their brain health [1]. Those with a diagnosis of dementia should be empowered to access interventions that can maximize independence and everyday function, improve quality of life, and prevent excess disability. Cognitive rehabilitation is an evidence-based intervention for supporting PLwD to achieve their goals by maintaining or enhancing cognitive and everyday function [2,3]. Although variations in approaches to general cognitive rehabilitation exist (e.g.,[4]), we refer to Linda Clare's Goal-Oriented Cognitive Rehabilitation (GREAT-CR) [5]. Research has shown that GREAT-CR has a positive impact on the lives of PLwD, and is a cost-effective intervention [6]. Evidence from single-case [7] and small-group intervention studies, along with randomized controlled trials (RCT) [5,8,9] indicates that GREAT-CR holds promise in enhancing goal attainment, well-being, and engagement in daily activities among individuals with mild cognitive impairment (MCI) and early-stage dementia [10]. GREAT-CR also shows potential in enhancing caregivers' self-reported quality of life [11]. The National Institute for Health and Clinical Excellence Guidelines [12], the World Alzheimer Report [13] and the Health Service Executive's (HSE) Model of Care for Dementia in Ireland [14] have recommended that GREAT-CR should be available to people with a diagnosis of dementia.

Irish PLwD have been advocating for their right to access early evidence-based interventions for many years. In 2013, a roundtable discussion with PLwD and their family members highlighted the importance of accessing early interventions that help the person with dementia to remain at home for longer [15]. Subsequently, the National Dementia Strategy [16] identified timely diagnosis and intervention as a "priority action area". Despite this, over ten years later there remains a treatment gap where evidence-based early interventions do not routinely follow diagnosis. Barriers to the implementation of psychosocial interventions in healthcare settings often include limited staff time and capacity, meaning that memory services may be motivated to deliver GREAT-CR but may lack the capacity to do so.

Our project aims to address implementation barriers by offering an innovative solution to resourcing issues. We aim to create a new service model for the delivery of community-based GREAT-CR, through a supervised placement program for early-career psychologists. In Ireland, masters/doctoral level psychologists are required to

complete supervised clinical placements as part of their training, but the demand for placements in older adult services is often unmet. We aim to alleviate this unmet need by creating a placement model within Irish Memory Clinics where trainee psychologists receive training and supervision to deliver GREAT-CR to PLwD in the community.

Our research proposes a two-phase approach to evaluating this implementation model. First, we will evaluate the impact of GREAT-CR on goal attainment, quality of life and cognitive function for PLwD when the intervention is delivered using a supervised placement model. Although strong evidence already exists for the efficacy of GREAT-CR, this is the first time that GREAT-CR will be evaluated in the context of trainee placements, and the first time a randomized N-of-1 design will be employed in a GREAT-CR study. We will determine whether goal attainment can be achieved as demonstrated in prior research when using our proposed delivery model. The intervention will be offered to approximately thirty-six PLwD, with outcomes evaluated at baseline, post-intervention and follow-up, and will include an embedded randomized N-of-1 single-case experimental design (SCED). N-of-1 or SCEDs a feature prominently in intervention literature [17]. Well-designed SCEDs can establish cause-effect relationships between interventions and outcomes [18] and are methodologically rigorous in evaluating intervention approaches [19]. The Oxford Centre for Evidence-Based Medicine ranks randomized N-of-1 designs as providing Level 1 evidence for intervention effectiveness [20], similar to reviews of RCTs [17]. This experimental method is recognized as especially important for optimizing personalized intervention approaches [21].

The second phase of the research will assess the effectiveness of the implementation model using an implementation-science approach. Implementation science is focused on identifying factors that impact on the uptake of interventions as part of routine care, rather than focusing on the effectiveness of the intervention itself, with the aim to enhance the availability and public health impact of evidence-based interventions [22]. The strategy for the proposed pilot is developed using the GREAT-iP (Goal-oriented Cognitive Rehabilitation in Early-stage Alzheimer's and related dementias: a multi-center single-blind randomized controlled Trial into Practice) implementation framework [23]. This was designed to support widespread implementation of CR into service provision for people with early-stage dementia. An implementation study involving 14 UK-based organizations demonstrated that this framework is effective in supporting the delivery of CR in routine services [23]. Qualitative and quantitative data will be collected at organizational, practitioner, and service-user level to evaluate the feasibility, acceptability, appropriateness, and potential sustainability of the implementation model. The design is informed by recommendations for mixed and qualitative methodologies in implementation science [22], which suggest that deductive, positivist approaches, guided by the implementation framework, may be preferable to constructivist, exploratory, inductive paradigms [24]. Semi-structured interviews will be conducted with a sample of key stakeholders involved in the pilot study (see methods section). Framework analysis [25] will inform development of interview guides, data collection, and analysis. Quantitative indicators of implementation success will be derived by assessing outcomes against specified targets and analyzed descriptively to support the qualitative findings.

## Research aims

This pilot study addresses three key aims: (1) to investigate whether a supervised placement model can be effectively utilized within memory services to enhance the availability of GREAT-CR; (2) to explore the feasibility and acceptability of the proposed intervention delivery model for individuals with dementia, their families, trainee practitioners, and service providers; and (3) to examine the impact of GREAT-CR on goal attainment, cognitive function, and quality of life within the context of the supervised placements.

## Methods

The proposed research was approved by the St James's Hospital and Tallaght University Hospital Joint Research Ethics Committee on the 27th of May 2024. Reference number 2024-May-35703570. The protocol adheres to the Spirit guidelines [26,27]: see S1 Appendix. This pilot research involves two sub studies with five groups of research participants. Group 1,

participants with dementia, will take part in both sub-study 1 and sub-study 2, while the remaining four groups will participate in sub-study 2. See Fig 1 for sub-study methodologies and participant groups.

## Ethics declarations

Approval for all project research activities was obtained from the ethics committees at St James' Hospital and Tallaght University Hospital Joint Research Ethics Committee. All data from participants will be stored in accordance with National College of Ireland and Tallaght University Hospital data protection and retention policies. For both sub-studies, participants will receive an information sheet and consent form at least one week prior to participation. The study will be explained in detail, including the phases of the intervention, the duration and number of sessions, as well as the measurements to be taken. Participants will also be informed about any potential risks associated with the study. Additionally, they will be provided with information regarding data protection, including how their data will be used, who will have access to it, how it will be stored, and their rights concerning their personal data. Participants will be informed that there will be no cost to them and that the study has received approval from the appropriate research ethics committee.

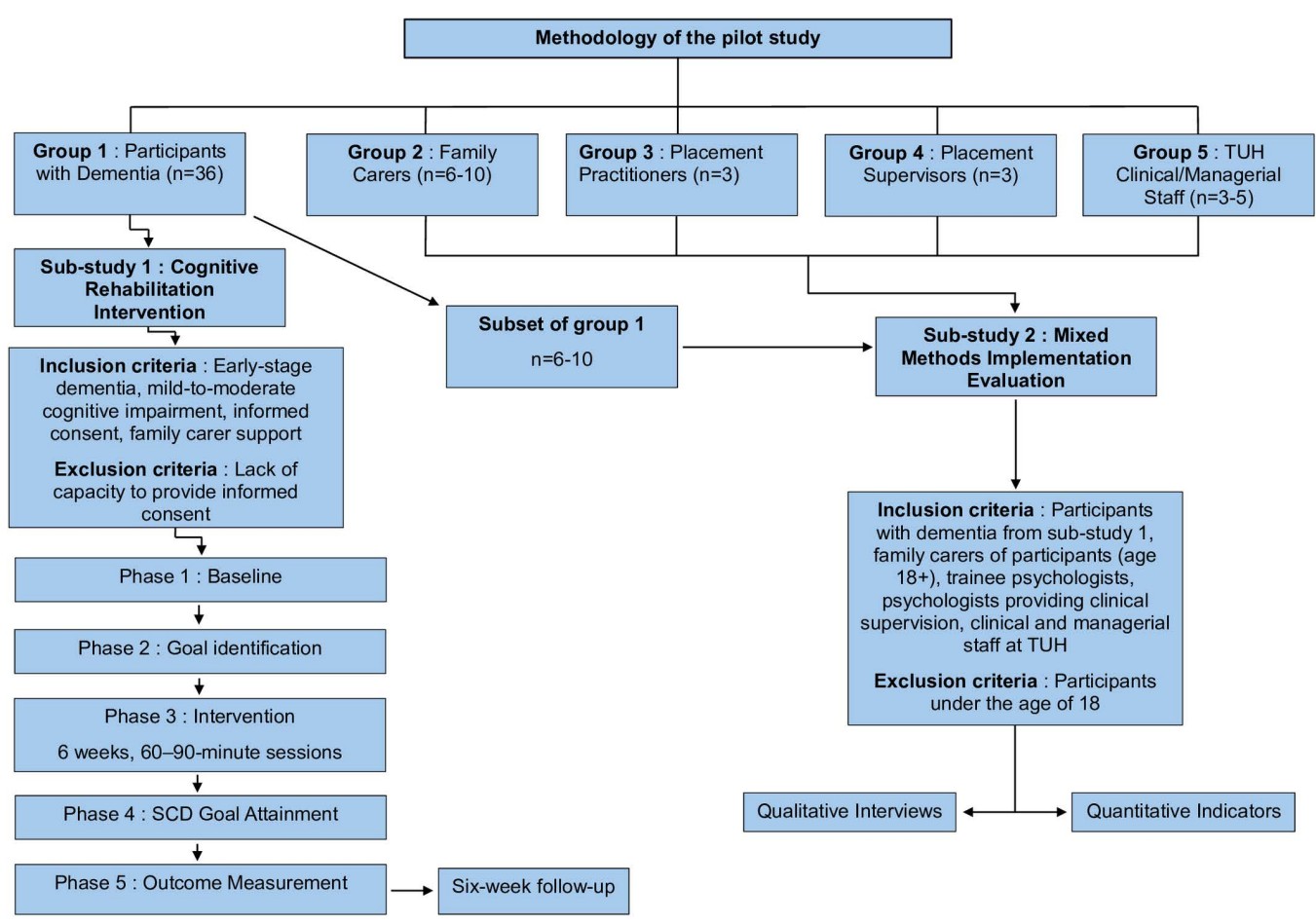

**Fig 1. Flowchart of the pilot study protocol.**

## Sub-study 1 – Cognitive rehabilitation intervention

### Participants

This study will include 36 participants with dementia, recruited through the Regional Specialist Memory Clinic (RSMC) at Tallaght University Hospital in Dublin, Ireland. Inclusion criteria specify that participants must have a diagnosis of early-stage dementia, exhibit mild-to-moderate cognitive impairment, be able to provide informed consent, and have a carer willing to provide support. Characteristics associated with positive outcomes in GREAT-CR trials [6] will also be considered for recruitment. These characteristics include being recently diagnosed, able to engage in daily activities and converse about them, motivated to reduce the impact of cognitive impairment on daily life, reasonably active, and supported by a carer who can encourage practice between sessions. Participants lacking the capacity to provide informed consent will be excluded from the research.

### Design and materials for sub-study 1

A quantitative pre-post design with and an embedded randomized *N*-of-1 SCED will be used. For all thirty-six participants, data on the primary outcome of goal attainment and satisfaction, and secondary outcomes of quality of life and cognition will be gathered at baseline, post intervention, and at a six-week follow-up. A subset of three participants will be randomly selected for the SCED analysis to evaluate the impact of GREAT-CR on objective measures of goal attainment. The SCED and reporting will adhere to the What Works Clearinghouse (WWC) Technical Documentation for SCDs [28] and the Single-Case Reporting Guideline in Behavioral Interventions [29]. A randomized multiple-baseline design will determine whether a causal relationship exists between the GREAT-CR intervention (independent variable) and the primary outcome of goal attainment (dependent variable). A multiple-baseline-across-participants approach will be utilized, where behavior on a specified goal will be assessed across three separate participants (e.g., [7]). We include both within-case (phase) randomization and start-point randomization [30]. As per WWC guidelines, an effect replication will be assessed across three participants, with three to five data points per phase (baseline and intervention). Interobserver agreement (IOA) will be collected by two independent assessors for each dependent variable.

### Measures/scales

Demographics and functional ability will be assessed at baseline. Demographic data will include age, gender, marital status, living status, level of education, and family carer demographics (see Table 1). Functional ability will be measured using the Functional Activities Questionnaire (FAQ) [31]. The FAQ evaluates participant's ability to perform ten instrumental activities of daily living, for example "Writing checks, paying bills, balancing a cheque book", which is indicative of functional independence in individuals with cognitive impairment. Six response options range from "Normal – I can do this by myself without difficulty" to "I need someone else to do this activity for me". Responses are scored from zero to three with a higher score indicating lower levels of independent functioning.

Our primary outcomes of interest, goal attainment and satisfaction, will be assessed using the Bangor Goal Setting Interview (BGSI) [11]. This semi-structured interview-based tool is designed to set and evaluate personalized goals for individuals with cognitive impairment [2]. Once goals have been identified, the PLwD's current level of goal attainment will be assessed on a 10-point Likert scale from 1 "cannot do or am not doing successfully" to 10 "can do and am doing very successfully" and is rated by the participant, the carer and the practitioner. Participants also rate their satisfaction with their current level of goal attainment on a 10-point Likert scale from 1 "extremely dissatisfied" to 10 "extremely satisfied". These outcomes will be assessed at baseline, post-intervention and at follow-up.

Secondary outcomes of cognitive function and quality of life will be assessed using the Montreal Cognitive Assessment (MoCA) [32] and the Quality of Life in Alzheimer's Disease (QoL-AD) [33], and will be assessed at baseline, post-intervention and at follow-up (see Table 1). The MoCA is a brief screening instrument for dementia used with individuals who may

**Table 1. List of outcomes, measures and timepoints at which associated measures will be utilized.**

| Outcome | Measure | Timepoint |
|---|---|---|
| Participant Characteristics | | |
| Demographic information | PLwD demographics: Age, gender, marital status, living status, level of education, self-rated health, time since dementia diagnosis, type of dementia. | Baseline |
| | Family carer demographics: Age, gender, level of education, relationship to PLwD, does the carer live with the participant (Y/N), average number of hours per week that care is provided to the participant. | |
| Functional ability | Functional Activities Questionnaire (self and carer rated) | Baseline |
| Primary Outcome | | |
| Goal Attainment and Satisfaction (Subjective) | Bangor Goal Setting Interview (BGSI): Ratings (participant, family carer, practitioner) of current functioning related to each goal (1 – unable/not currently doing to 10 – able to do well with no difficulty) and current level of satisfaction with goal attainment (1- extremely dissatisfied to 10-extremely satisfied). | Baseline Post-Intervention Follow-Up |
| Goal Attainment (Objective)* | Practitioner-recorded proportion of correct responses/percentage goal attainment when practicing selected goal (see Kelly et al., 2019 for an example). | Baseline Each intervention session Post-intervention Follow-up |
| Secondary Outcomes | | |
| Cognitive Function | Montreal Cognitive Assessment (MoCA) | Baseline Post-Intervention Follow Up |
| Quality of Life | Quality of Life in Alzheimer's Disease (QoL-AD) | Baseline Post-Intervention Follow Up |

*Measure will be collected only for SCD participants (n=3)

experience cognitive difficulties [34]. The MoCA assesses cognitive domains comprising visuospatial/executive, naming, memory, attention, language, abstraction and orientation. The test has a maximum score of 30. A cut-off score of 26 is recommended by the test developers with a score below 26 being considered indicative of cognitive impairment; one point is added to the total score if the person has 12 years or less of formal education [32]. It has been previously suggested that this cut-off score of 26 has poor specificity meaning there may be a risk of misclassifying cognitively able people as having impairment [35]. In the Irish context, population-specific norms have been developed which are stratified according to age and education level which combats this criticism [36]. These normative data will be used in the current study rather than the cut-off score recommended by the test developers.

The QoL-AD is a self and proxy-rater measure that assesses quality of life (e.g., physical health, energy, mood, living situation, memory, family, marriage, friends, etc.) [33]. The QoL-AD is composed of 13 questions, with a factorial structure comprising three dimensions: physical, social, and psychological well-being. The total score ranges from 13 to 52, with each question rated on a 4-point Likert scale ranging from "Poor" to "Excellent". A high score indicates a higher quality of life [33,37].

## Procedure

The intervention study comprises three phases: (1) pre-intervention (baseline) assessments and goal identification, (2) intervention phase, and (3) post intervention assessments.

**Phase 1: Pre-intervention (Baseline) assessments and goal identification.** The first visit from the placement practitioner will provide an opportunity for the participant to ask questions, and for the practitioner to explain the study procedures and expectations again. If the participant agrees to continue, written informed consent will be obtained. The practitioner will gather baseline data (see Table 1 for list of baseline measures) and discuss possible goals. The first session would be expected to last around 90 minutes. Goal identification can take some time so may occur during the initial visit or in a subsequent visit. The placement practitioner will assist participants and their caregivers in identifying

up to three personalized rehabilitation goals using the BGSI. Once goals are identified, the practitioners will work with the participant to operationally define the goals to ensure they are specific, measurable, achievable, relevant and time-oriented (SMART). Practitioners will subsequently work with their clinical supervisors to develop the intervention and session plans, and to determine how best to measure goal performance within sessions.

For the embedded N-of-1 design, baseline data on goal performance will be objectively measured (e.g., proportion of correct responses on a face-name recall goal) with baseline data monitored across three participants concurrently. The participants will be randomly assigned to different baseline durations (see Table 2 for an example). This allows for causal inferences by applying the intervention sequentially to each. An effect replication will be assessed across participants with at least 3–5 data-points per phase.

**Phase 2: Intervention.**  The intervention will be tailored to each participant's goals, utilizing techniques from the cognitive rehabilitation literature. The intervention will span approximately six to eight weeks with weekly 60–90-minute sessions, conducted at the participant's home unless otherwise preferred (option available to conduct sessions in the Memory Clinic). The end of each session will include a review with the caregiver and planning for home practice. Rehabilitative approaches may include compensatory or restorative strategies. Compensatory strategies may include environmental or task modifications, practicing the use of memory aids like diaries, calendars, or medication reminders, or other techniques tailored to the participant's needs like relaxation. Restorative strategies to support new learning or relearning of functional skills/information may incorporate task analysis, chaining, prompting and fading, multiple exemplar training, mnemonics, errorless learning and/or spaced retrieval. Strategies used during sessions will be documented weekly, and participants will receive a summary report at the end of the intervention. Practitioners will monitor participant's progress on selected goals during each session to ensure intervention efficacy. For the N-of-1 participants, the intervention start-point will be predetermined (see Table 2) for their goals. The intervention is due to span six to eight weeks but for goals with longer baseline phases this may be less. Intervention length will be determined by participant need, availability and consent to work on goals.

**Phase 3: Post-intervention outcome measurement.**  Standardized outcome measures will be collected to assess the following: the primary outcome, goal attainment and satisfaction (BGSI); and the secondary outcomes, quality of life (QoL-AD) and cognitive function (MoCA). A follow-up meeting will then be arranged for 6-weeks post intervention to collect the final set of outcome measures. Outcome measures will be collected using an assessment booklet, which includes all relevant measures for each time point. The research coordinator will verbally administer the questionnaires and record all responses in the paper assessment booklets.

## Proposed analytic approach

Descriptive statistics will be calculated to compare scores on primary and secondary outcomes at baseline, post-intervention, and at six-week follow-up for all participants who receive CR (n=36). For the embedded *N*-of-1 study, visual analysis will objectively examine the level, trend, and variability of data within each phase, as well as the immediacy of

**Table 2.  Sample plan for the N-of-1 design showing random allocation of participants to predetermined baseline lengths and random allocation of intervention start-points from a possible two.**

| Participant # (randomly allocated) | Possible # baseline data points | Possible intervention start points | Actual design – *determined by ExPRT start point randomizer* |
|---|---|---|---|
| 3 | 3/4 | 4/5 | AAAABBBBBBBB |
| 1 | 5/6 | 6/7 | AAAAABBBBBBB |
| 2 | 7/8 | 8/9 | AAAAAAABBBBB |

effect and percentage of non-overlapping data. If evidence of an intervention effect is observed, effect size estimates will be calculated. Then, ExPRT (Excel Package of Randomization Tests) software will be used to analyze the randomized single-case multiple-baseline data, providing graphical representation and effect size estimates [38]. The design includes three multiple-baseline design tiers and two potential intervention start-points per tier. The two most extreme mean phase-difference outcomes in the predicted direction will yield a statistically significant result at the .05 level [30].

## Sub-study 2 – mixed methods implementation evaluation

### Participants

In sub-study 2, purposive sample of key stakeholders will include a subset of participants with dementia from group 1 in the sub-study 1 (n= 6–10) and family carers (n= 6–10) engaged with the intervention, placement practitioners (n= 3–5), placement supervisors (n= 3), and clinical and managerial staff at the pilot site (n=3–5). Participants with dementia who have taken part in the CR intervention as part of sub-study 1 and family carers of these participants (aged 18 and above) will be offered as opportunity to take part in the interviews. Also, trainee psychologists who have completed the 9-month clinical placement at the RSMC as part of this study will be interviewed. Psychologists providing clinical supervision to these trainees, specifically members of the project team, will also be included. Lastly, clinical and managerial staff at the memory clinic involved in the delivery of services to people with dementia will be part of the qualitative interviews. Participants under the age of 18 will be excluded from sub-study 2.

### Design and materials of sub-study 2

This sub-study 2 will employ a mixed-methods implementation evaluation to assess the feasibility, acceptability, appropriateness, and potential sustainability of the implementation model. Both qualitative and quantitative data will be collected at the organizational, practitioner, and service-user levels. Semi-structured interviews will be conducted with stakeholders described above. Framework analysis [25] will inform development of the interview guides, data collection, and analysis. The analyst will familiarize themselves with the transcripts, and data will be systematically coded using a matrix to summarize data by case and by code. Comparison of codes across and within cases will be used to identify and refine themes using a combination of inductive and deductive processes; themes will be mapped to pre-defined constructs proposed to influence implementation (identified from relevant implementation frameworks) while also allowing scope for unanticipated themes to be identified inductively. Following guidelines for framework analysis, two researchers will independently code the first few transcripts; the research team will then review initial codebooks to agree the analytical framework (set of codes) that will be applied to the remaining interviews. An iterative process will be used to review codes, themes, and findings, with any differences of opinion regarding the interpretation of findings resolved through discussion by the research team.

Quantitative indicators of implementation success will be derived by assessing outcomes against specified targets and analyzed descriptively to support the qualitative findings related to each implementation outcome. The key implementation outcomes selected for evaluation are outlined in Table 3.

### Procedure

**Groups 1 and 2: Participants with dementia who received CR intervention and family members/caregivers.** A subset of 6–10 participants with dementia from sub-study 1 and a subset of 6–10 family members/caregivers supporting participants in sub-study 1 will be invited for qualitative interviews to evaluate the implementation. Interviews, conducted by the research coordinator, will focus on participants' and family members' experiences with the CR intervention. Interviews will be semi-structured, based on existing literature and frameworks, lasting 30–60 minutes, and will be recorded and transcribed for analysis.

**Groups 3, 4, and 5: Placement practitioners, supervisors, and memory clinic staff.** A subset of placement practitioners (n = 3–5), supervisors (n = 3), and a purposive sample of clinical/managerial staff at the pilot site (n = 3–5) will be recruited for qualitative interviews. The research coordinator will conduct these interviews, focusing on the training, support, and experiences related to delivering the intervention, as well as barriers, facilitators, and sustainability of the implementation. Semi-structured interviews will last 30–60 minutes and will be recorded and transcribed for analysis.

## Information power

Sample size in qualitative research is a contentious issue; the concept of data saturation is commonly applied but the utility of this approach has been questioned [39]. Recent guidance for qualitative implementation research suggests that sampling should primarily focus on the depth and quality of the data provided rather than the quantity of participants. This aligns with the concept of "information power" for qualitative sampling, which proposes that the more information the sample holds (relevant to the research question) the fewer participants are needed [40]. The sample for the current study is likely to have high information power, considering the specific aims of the study and the inclusion of key stakeholders who are most critical to the implementation effort. The sample size is also consistent with previous qualitative implementation studies.

## Discussion

### Summary of expected outcomes

This study is expected to yield several outcomes. Firstly, it aims to provide further evidence on the impact of GREAT-CR on goal attainment and satisfaction, quality of life and cognitive function for individuals with dementia. By demonstrating the benefits of GREAT-CR, the study evidence can provide a stronger rationale to support its inclusion as a standard intervention in dementia care. Moreover, participants with early-stage dementia will gain direct access to evidence-based GREAT-CR interventions, thereby enhancing their treatment and potentially delaying the progression of cognitive decline. This direct access is particularly important in offering timely support to those in the early stages of the condition. A new service delivery model for memory clinics is another anticipated outcome. The successful implementation of this model could transform current practices, making GREAT-CR more widely available and integrated into routine care. This model will also serve as a blueprint for other memory clinics aiming to adopt similar approaches. The study will also increase the availability of placements for trainee psychologists within dementia services. By participating in this project, trainee

Table 3. Sub study 2 implementation outcomes.

| Outcome | Outcome Definition (12) | Research Question |
|---|---|---|
| Feasibility | Extent to which the innovation can be practically used in a given setting | Can a supervised placement model be utilised in memory services to increase availability of CR for people with early-stage dementia? |
| Acceptability | View among stakeholders that the given innovation is agreeable or satisfactory | Is the proposed model of intervention delivery acceptable for people with dementia, their families, trainees, and service providers? |
| Appropriate-ness | Perceived compatibility with needs and practices of a setting or population; perceived utility in addressing a given problem | Is CR impactful for people with dementia; does it improve functional outcomes? |
| | | Does the implementation model effectively address the needs of the stakeholders? |
| (Potential) Sustainability | Extent to which the innovation is maintained or routinised within a setting over time | Can the placement model be managed effectively within the memory services? |
| | | Do clinical and managerial staff intend to maintain the intervention and placement model going forward? |
| | | What barriers and facilitators affect the implementation of CR using the supervised placement model? |

psychologists will gain experience in delivering GREAT-CR, which will enhance their clinical skills and expertise in working with dementia patients. Lastly, the project is expected to expand the availability of GREAT-CR in Irish memory clinics. By training more clinicians to deliver GREAT-CR, the study will help ensure that a greater number of out-patients can benefit from this intervention, thereby enhancing the overall quality of dementia care provided in national memory clinics.

## Anticipated implications

The direct impact of this study on individuals with dementia, particularly those in the early stages, is expected to be significant. Access to GREAT-CR is envisaged to empower PLwD to address their identified goals in a meaningful way, hopefully leading to improvements in their daily lives. By enhancing practical strategies for everyday challenges, individuals with dementia can maintain independence and community living for longer, thereby reducing the risk of excess disability, and premature long-term care placement. Family members of individuals with dementia might also benefit from the study's outcomes. Access to GREAT-CR in the home environment can not only alleviate concerns regarding independent functioning but also provide practical solutions that can be implemented in real-life settings, enhancing the meaningfulness of interventions for both the individual and their family.

Furthermore, the study's impact extends to broader dementia care service provision. By piloting and evaluating an early-intervention framework, the study aims to provide evidence for a cost-effective and scalable solution to increase the availability of early dementia interventions. If proven effective, the placement model could serve as a roadmap for scaling up the delivery of CR across the country, thus addressing priorities outlined in Ireland's Model of Care for Dementia and meeting the needs of individuals with dementia. Establishing a collaborative network between key stakeholders, including the Alzheimer Society of Ireland, clinical memory services, universities offering psychology training, and individuals with dementia, will further enhance the study's impact. This network can facilitate the expansion of CR delivery to additional clinical sites, training providers, and stakeholders, ensuring wider access to early interventions for people with dementia. In summary, this study aims to develop and evaluate a new model for providing early interventions that does not exist yet, bridging the gap between research, policy, and practice with an immediate impact. The anticipated societal and economic benefits align with national priorities and the needs of individuals with dementia. The study is expected to conclude by autumn 2026. Based on the findings from the study, guidelines for the delivery of GREAT-CR through a clinical placement model will be developed. These guidelines will facilitate and promote the scaling up of this new service model across the country.

## Standardization and data quality

The psychology trainees will be required to participate in several training sessions during which all intervention strategies, outcome measures and assessment materials will be discussed, demonstrated, and practiced. Participants will be assured that all information provided will be treated in strictest confidence, and data will undergo anonymization procedures. Data storage will adhere strictly to the data protection and retention policies outlined by the National College of Ireland and Tallaght University Hospital. Anonymized datasets will be archived in an open data repository.

## Supporting information

**S1 Appendix. Completed SPIRIT checklist.**
(DOCX)

## Author contributions

**Conceptualization:** Caoimhe Hannigan, Garret McDermott, Michelle Kelly.

**Funding acquisition:** Caoimhe Hannigan, Garret McDermott, Helena Lydon, Sean P Kennelly, Michelle Kelly.

**Investigation:** Caoimhe Hannigan, Garret McDermott, Michelle Kelly.

**Methodology:** Caoimhe Hannigan, Michelle Kelly.

**Project administration:** Caoimhe Hannigan, Garret McDermott, Michelle Kelly.

**Supervision:** Caoimhe Hannigan, Garret McDermott, Helena Lydon, Sean P Kennelly, Michelle Kelly.

**Writing – original draft:** Caoimhe Hannigan, Antoine Lemercier, Michelle Kelly.

**Writing – review & editing:** Antoine Lemercier, Garret McDermott, Helena Lydon, Sean P Kennelly, Michelle Kelly.

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
