## [Decision Letter · Decision Letter 0]

18 Mar 2025

Study protocol for an observational research study with an embedded N-of-1 design: Increasing the availability of goal-oriented cognitive rehabilitation for people living with dementia in Ireland

PONE-D-24-37236

Dear Dr. Lydon,

We’re pleased to inform you that your manuscript has been judged scientifically suitable for publication and will be formally accepted for publication once it meets all outstanding technical requirements.

Kind regards,

Ioannis Liampas, MD. PhD

Academic Editor

PLOS ONE

Journal Requirements:

1. Thank you for stating the following financial disclosure:

“The current research is funded by the Alzheimer Society of Ireland under the Dementia Awards Scheme.”

Please respond by return e-mail so that we can amend your financial disclosure and competing interests on your behalf.

Additional Editor Comments (optional):

Reviewers' comments:

Reviewer's Responses to Questions

**Comments to the Author**

1. Does the manuscript provide a valid rationale for the proposed study, with clearly identified and justified research questions?

Reviewer #1: Yes

Reviewer #2: Yes

2. Is the protocol technically sound and planned in a manner that will lead to a meaningful outcome and allow testing the stated hypotheses?

Reviewer #1: Yes

Reviewer #2: Yes

3. Is the methodology feasible and described in sufficient detail to allow the work to be replicable?

Reviewer #1: Yes

Reviewer #2: Yes

4. Have the authors described where all data underlying the findings will be made available when the study is complete?

Reviewer #1: Yes

Reviewer #2: Yes

5. Is the manuscript presented in an intelligible fashion and written in standard English?

Reviewer #1: Yes

Reviewer #2: Yes

6. Review Comments to the Author

You may also provide optional suggestions and comments to authors that they might find helpful in planning their study.

Reviewer #1: The study objectives and methods are well planned, comprehensive and evidence-based. Provided the results/analysis of the study is positive/encouraging, similar approach can be applied to resource-limited settings or with diseased specific population with mild cognitive impairment. I suggest submission of result manuscript be submitted at this journal too.

Reviewer #2: The research design employed small sample approach to intervene early-stage dementia patients, and tried to establish a new practice model for students.

Although single-case experimental design is to be used, the research methodology is comprehensive and could make causal inferences. The new model can supply with practical component in traditional teaching, enhancing clinical practice of psychological interns. The research design can achieve the experimental objective and has practical applicational value.

The authors need to ensure experimental methods to guarantee the validity of causal inference in the study. For example, conducting measurements from multiple sources, randomized multiple baselines, etc.

In subsequent statistical analysis, the authors may consider using statistical inference methods for small samples, such as Bayesian approaches.

7. PLOS authors have the option to publish the peer review history of their article (what does this mean? ). If published, this will include your full peer review and any attached files.

**Do you want your identity to be public for this peer review?** For information about this choice, including consent withdrawal, please see our Privacy Policy .

Reviewer #1: **Yes: ** Norhamizan Hamzah

Reviewer #2: No

---

## [Editor Report · Acceptance letter]

PONE-D-24-37236

PLOS ONE

Dear Dr. Lydon,

I'm pleased to inform you that your manuscript has been deemed suitable for publication in PLOS ONE. Congratulations! Your manuscript is now being handed over to our production team.

Kind regards,

on behalf of

Dr. Ioannis Liampas

Academic Editor

PLOS ONE